# Gate-tunable large magnetoresistance in an all-semiconductor spin valve device

M. Oltscher[1], F. Eberle[1], T. Kuczmik[1], A. Bayer[1], D. Schuh[1], D. Bougeard[1], M. Ciorga[1] & D. Weiss[1]

A large spin-dependent and electric field-tunable magnetoresistance of a two-dimensional electron system is a key ingredient for the realization of many novel concepts for spin-based electronic devices. The low magnetoresistance observed during the last few decades in devices with lateral semiconducting transport channels between ferromagnetic source and drain contacts has been the main obstacle for realizing spin field effect transistor proposals. Here, we show both a large two-terminal magnetoresistance in a lateral spin valve device with a two-dimensional channel, with up to 80% resistance change, and tunability of the magnetoresistance by an electric gate. The enhanced magnetoresistance is due to finite electric field effects at the contact interface, which boost spin-to-charge conversion. The gating scheme that we use is based on switching between uni- and bidirectional spin diffusion, without resorting to spin–orbit coupling. Therefore, it can also be employed in materials with low spin–orbit coupling.

[1] Institute for Experimental and Applied Physics, University of Regensburg, 93055 Regensburg, Germany. Correspondence and requests for materials should be addressed to M.C. (email: mariusz.ciorga@ur.de)

Since the original spin field effect transistor (sFET) proposal[1], spin injection into nonmagnetic semiconductors (SC), subsequent spin manipulation, and conversion of spin information into large electric signals have become the main challenges of spintronics[2, 3]. It has been established that efficient spin injection from a ferromagnet (FM) into a semiconducting channel requires a tunnel barrier[4, 5], which has been demonstrated in experiments on various FM/SC systems[6–11]. In some cases, the measured spin signals have been even larger than the values predicted by the standard diffusive model of spin injection[8, 10, 11]. Large spin accumulation in a SC does not guarantee, however, a large magnetoresistance ratio $MR = \Delta R / R^P$, with $\Delta R = R^{AP} - R^P$, and $R^{P(AP)}$ being the two-terminal (2T) resistance of the device, measured between the ferromagnetic contacts when magnetizations of the contacts are either parallel (P) or antiparallel (AP) to each other. A sizeable magnetoresistance (MR), being indispensable for novel spin-transistor concepts[1, 12], requires that the electrons' dwell time in the transport channel is much smaller than the corresponding spin relaxation time[13–15]. This condition is particularly difficult to satisfy in semiconductors, as a high-interface resistance needed for efficient injection increases both the dwell time and the spin-independent contribution to $R^{P(AP)}$. This is the

reason for small $MR < 1\%$ obtained so far for semiconductor channels[14, 16–18].

One aspect that has not been fully exploited in semiconductor spin devices, though, is how an electric field in the channel and particularly in the contacts affects the MR. Albeit the influence of electric fields on both spin transport and spin-to-charge conversion in semiconducting channels has been discussed before[19, 20], until very recently[21], the experiments have been typically limited to cases involving only single-biased FM/SC junctions[22–25]. The standard two-terminal configuration implies, however, that two FM/SC contacts are biased in opposite directions. Therefore, a nonrectifying $I–V$ characteristic of the contacts is needed to drive a sufficiently large current through the device, which generates a sizeable spin accumulation in the channel and allows taking advantage of electric field effects.

While large MR is a crucial prerequisite for operational spintronic devices, their most desired functionality is the efficient electrical control of $R$ via control of the spin state. In the seminal proposal of the sFET[1] and in related experiments[26–29], spin–orbit fields mediate sFET action. The realization of this concept requires materials with spin–orbit coupling large enough for efficient rotation of traveling spins but small enough to prevent spin relaxation between source (S) and drain (D). For low

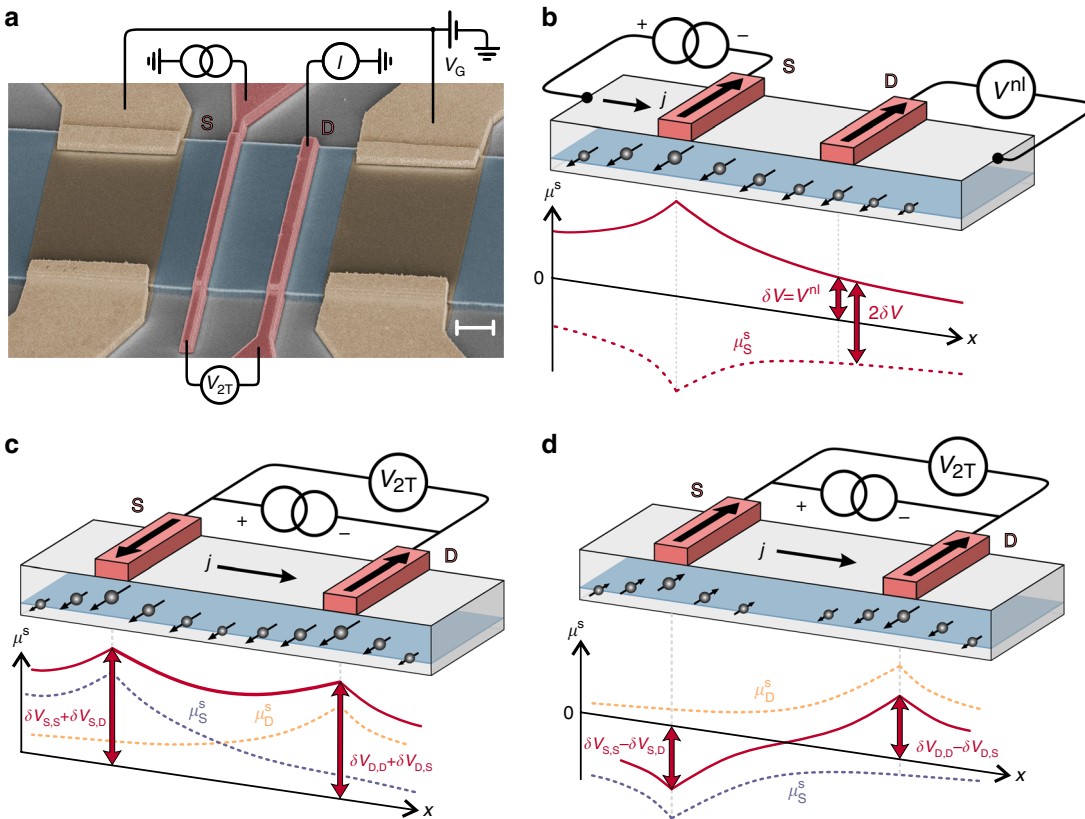

**Fig. 1** Devices and measurement configurations. **a** Colored SEM image of one of the gated devices. Two narrow contacts in the middle, 500 and 700 nm wide, separated by 3.6 μm, are the ferromagnetic source (S) and drain (D) leads. Two electrostatic gates are used to confine the spins between S and D for $V_G < 0$. Additionally, the electric circuit for 2T measurements is sketched. The scale bar length is 2 μm. **b** Nonlocal configuration, with S being an injector and D a detector. The majority spins are injected into the channel from the negatively biased S and diffuse along the channel. The solid line indicates the spin accumulation profile $\mu^s(x) = \frac{1}{2}(\mu_\uparrow(x) - \mu_\downarrow(x))$, where $\mu_{\uparrow(\downarrow)}$ is the quasichemical potential for the corresponding spin direction. The dashed line shows $\mu^s(x)$ when either the polarity of the injection current $j$ or the magnetization direction of the source has been reversed. $\mu^s$ underneath the drain induces the spin-dependent voltage $\delta V$, measured nonlocally as $V^{nl}$, which changes by $\Delta V^{nl} = 2\delta V$ each time the magnetization of either of the contacts is reversed. **c**, **d** 2T configuration with a positively biased source and a negatively biased drain for antiparallel and parallel orientation of the magnetization of the contacts, respectively. Solid lines indicate the total spin accumulation $\mu^s(x) = \mu^s_S(x) + \mu^s_D(x)$, where $\mu^s_{S(D)}$ (dashed line) is the spin accumulation generated at the source (drain). In AP (P) configuration, both components have the same (opposite) sign. As a result, $\mu^s$ is larger (smaller) in the AP (P) configuration. The 2T voltage difference between both configurations is given by $\Delta V = 2\delta V_{S,D} + 2\delta V_{D,S}$, where $\delta V_{i,j}$ is the voltage drop at contact $i$ due to $\mu^s_j$

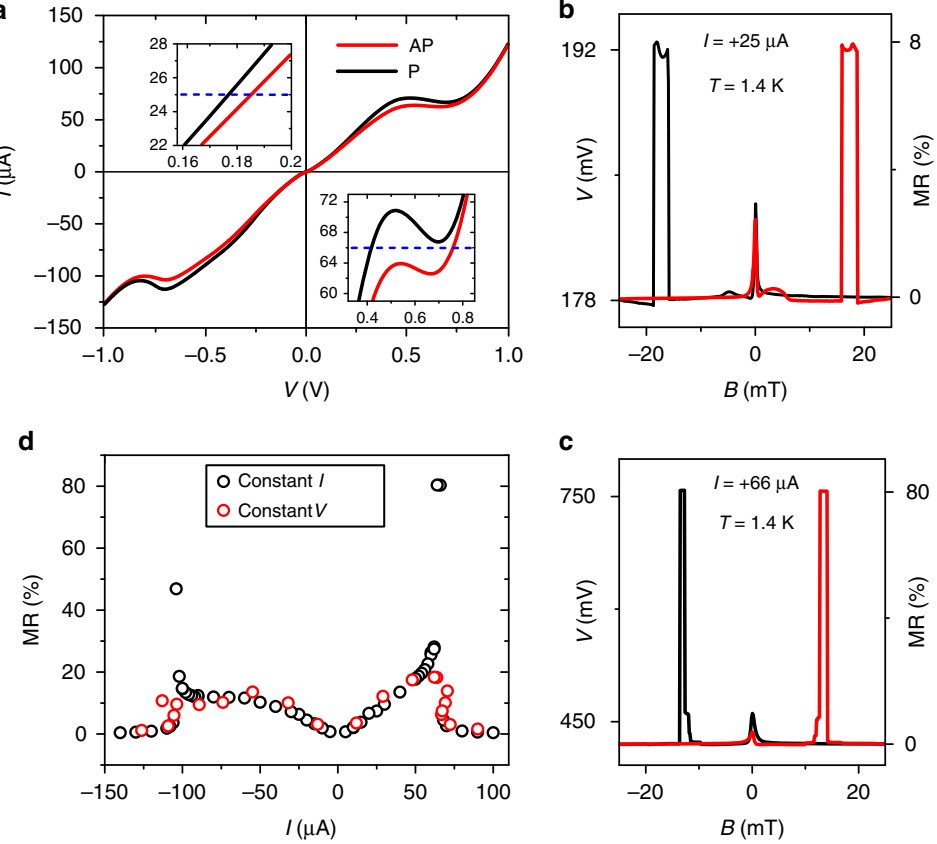

**Fig. 2** Magnetoresistance ratio and its bias dependence. **a** Two-terminal $I$–$V$ characteristics of the ungated device for P and AP orientations of source and drain magnetization. S–D separation $d = 3.6\,\mu\text{m}$. Insets show sections of the $I$–$V$ curve (same units as in the main graph) in the linear regime at low bias (top) and in the nonlinear region of negative differential resistance (NDR) of the Esaki diode (bottom). Blue dashed lines indicate current values at which spin valve traces, shown in **b**, **c**, were taken. **b**, **c** Two-terminal magnetoresistance of the device for $B$ swept along the contacts' axes, taken at a constant current of $I = +25\,\mu\text{A}$ and $+66\,\mu\text{A}$, respectively. Red (black) traces correspond to sweeping $B$ in a positive (negative) direction. Typical spin valve behavior is observed upon magnetization reversal, with the higher voltage level corresponding to AP magnetization orientation of source and drain contacts. The coercive fields of the contacts decrease for higher bias as a result of a local heating. Consistent behavior of the spin valve is observed at higher temperatures (Supplementary Note 8). **d** Magnetoresistance ratio MR $= \Delta R/R^{\text{P}}$ as a function of the applied current. The results of both measurement modes, with constant current (black circles) and constant voltage (red circles), are shown. In the latter case, the MR is plotted vs. the current in the parallel configuration. Similar MR behavior was observed on eight different devices

spin–orbit coupling (SOC), e.g., the corresponding spin precession frequency is low, what in turn calls for long transport channels. Therefore, alternative approaches have also been suggested, where transistor action is obtained either through electrical control of the spin injection process[12, 30] or of the spin transport efficiency along the channel[31–33].

In this paper, we present devices that meet both the requirement of a nonrectifying $I$–$V$ characteristic, leading to large MR signals, and offer an alternative way of tuning MR, without invoking SOC. Ferromagnetic source and drain of the transistor-like structures consist of (Ga,Mn)As/GaAs spin Esaki diodes[34, 35] with a nearly linear $I$–$V$ characteristic at low bias and very efficient spin injection from ferromagnetic (Ga,Mn)As into both bulk[9] and 2D GaAs channels[11] (Supplementary Note 1). Using a two-dimensional electron system (2DES) as a transport channel allows for additional control of spin accumulation in the channel through electric gates via SOC or, as used here, to confine spins in the region between the leads. This provides an extra knob for controlling the resistance of the device.

## Results

**Devices and their spin injection parameters.** In Fig. 1a, we show the SEM picture of one of the investigated devices, which were all fabricated from the same single heterostructure, as described in Methods. They consist of a lateral channel defined within the 2DES, with ferromagnetic spin Esaki diode contacts on top serving as injectors and detectors of spin accumulation. Two types of devices were investigated. The nongated ones, with multiple FM contacts (Supplementary Note 1), were used both for four-terminal nonlocal measurements (Fig. 1b) and for 2T local measurements (Fig. 1c, d). From nonlocal measurements on such devices, we determine spin transport parameters of the channel, like spin injection efficiency, spin diffusion length, and spin relaxation times[36]. The gated devices (Fig. 1a) feature a pair of FM contacts and a pair of electric gates used to confine spins in the region between ferromagnetic source and drain contacts. This type was used mainly for 2T local measurements to demonstrate electric-gate control of MR.

According to the commonly accepted standard model of spin injection[5, 13], the key parameters for efficient spin injection and large MR signals are effective spin resistances of the barrier $R_{\text{T}}$ and the channel $R_{\text{ch}}$. The maximum spin signal is expected when $R_{\text{ch}} \ll R_{\text{T}} \ll R_{\text{ch}}\lambda_{\text{s}}/d$, where $\lambda_{\text{s}}$ is the spin diffusion length and $d$ is the length of the channel between source and drain contacts. While the left condition is typically satisfied for our devices, the right condition is not fulfilled, as the ratio $R_{\text{T}}/R_{\text{ch}}$ is too large by

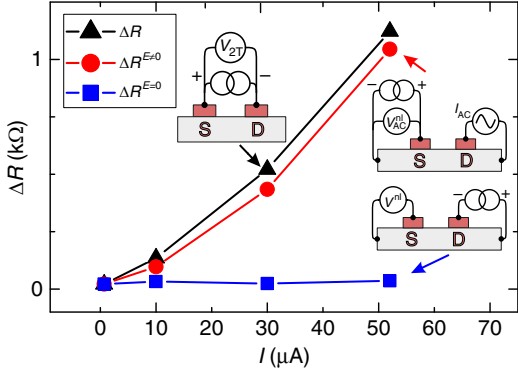

**Fig. 3** Influence of the electric field on the magnetoresistance. Comparison of the measured two-terminal spin signal $\Delta R$ (black triangles) with $\Delta R^{E=0} = \Delta R_{\mathrm{S,D}}^{\mathrm{nl}} + \Delta R_{\mathrm{D,S}}^{\mathrm{nl}}$ (blue squares), obtained when neglecting electric field effects, and with $\Delta R^{E\neq0} = \eta_{\mathrm{S}}\Delta R_{\mathrm{S,D}}^{\mathrm{nl}} + \eta_{\mathrm{D}}\Delta R_{\mathrm{D,S}}^{\mathrm{nl}}$ (red circles), obtained when taking drift in the channel and electric fields in the contacts into account. The results are shown for four different currents between a positively biased source and a negatively biased drain. $\Delta R_{\mathrm{S,D}}^{\mathrm{nl}}$ and $\Delta R_{\mathrm{D,S}}^{\mathrm{nl}}$ were obtained from nonlocal measurements with the unbiased detector and either a negatively biased drain (the corresponding sketch is shown) or a positively biased source as a spin injector, to match the polarity of the given lead in 2T measurements. Enhancement ($\eta_{\mathrm{S}}$) and suppression ($\eta_{\mathrm{D}}$) factors for positively biased S and negatively biased D, respectively, were obtained from nonlocal measurements with biased detectors (the sketch for the case of the biased source is shown). For details, see Supplementary Note 3. Solid lines are a guide to the eye

two orders of magnitude (Supplementary Note 1). This means that the dwell time of electrons in the channel is much larger than the spin relaxation time[13–15, 37]. It shows that the standard model of spin injection, without taking into account the electric field effects, does not explain the large MR signals that we observe in our devices, as we discuss in the following section.

**MR measurements.** First, we present in Fig. 2 the large MR signals measured in a nongated device. Figure 2a shows the corresponding I–V curves for AP (Fig. 1c) and P (Fig. 1d) configurations. The curves are almost symmetric with respect to $V = 0$, enabling us to pass currents up to 100 μA through the device. Clearly, for a given voltage V, the current is lower for the AP configuration, indicating a larger resistance due to an increased spin accumulation in the channel, as sketched in Fig. 1c, d. The spin origin of this resistance is confirmed by measurements in an out-of-plane magnetic field, which depolarizes the in-plane spins, thus reducing the measured value of $R^{\mathrm{P(AP)}}$ due to suppression of its spin-dependent contribution (Supplementary Note 7). The splitting between I–V curves for P and AP configurations increases with V, causing a larger MR for higher bias. This is confirmed by the corresponding measurements of the spin valve (SV) effect, where the magnetic field B is swept along the contacts' axis (see Methods). Figure 2b and c show typical SV signals measured in the linear (b) and in the nonlinear (c) regime of the Esaki diode. Regular SV behavior is observed as a result of different coercive fields of source and drain contacts, leading to switching between AP and P alignment of the magnetization in both contacts. At $B = 0$, the feature related to dynamic nuclear polarization effects[38] is also observed. The full bias dependence of the MR is plotted in Fig. 2d. In the linear regime, MR = $\Delta R/R^{\mathrm{P}}$ clearly increases with bias, reaching ~20%. In this regime, the MR is independent of the chosen measurement configuration, i.e., it shows the same values for constant current and constant voltage

measurements. In the nonlinear regime, however, the signal is further enhanced when the device is biased with a constant current. Then, the difference between the I–V characteristic's P and AP branch leads to a large voltage output $\Delta V = V^{\mathrm{AP}} - V^{\mathrm{P}}$. With $\Delta V \cong 350\,\mathrm{mV}$ measured for $I = +66\,\mu\mathrm{A}$, this results in a MR ratio of ~80%. A similar behavior was observed on eight different samples. The corresponding data for one of the gated devices are exemplarily shown in Supplementary Note 2.

**The role of electric fields.** To demonstrate the key role of electric fields in obtaining large MR, we compare our $\Delta R$ data with values expected from the standard model of spin injection[5, 13]. Following this model, a charge current I flowing across a FM/SC junction generates a spin accumulation $\mu^{\mathrm{s}}$ in the SC channel. On the other hand, the spin accumulation in the vicinity of such a ferromagnetic contact leads to a spin-dependent voltage that can be measured due to Silsbee–Johnson spin charge coupling[39]. In a 2T configuration, where charge current flows between source and drain, each FM contact serves both as injector and detector, i.e., a spin accumulation is created and detected at both contacts (Fig. 1c, d). The voltage drop at each FM/SC interface therefore contains two spin-dependent contributions, one from $\mu^{\mathrm{s}}$ generated at this particular contact and a second one from the spin accumulation generated at the other contact. In the AP configuration, these voltages add up, whereas in the P configuration, they partially cancel out. As a result, the voltage difference between P and AP configurations, i.e., the total SV signal of the 2T device, is given by $\Delta V = 2\delta V_{\mathrm{S,D}} + 2\delta V_{\mathrm{D,S}}$, where $\delta V_{\mathrm{i,j}}$ is the contribution of the spin accumulation generated at contact j that diffused to contact i. The different contributions to the SV signal $\Delta V$ are illustrated in Fig. 1c, d. $2\delta V_{\mathrm{i,j}}$ corresponds directly to the SV signal $\Delta V_{\mathrm{i,j}}^{\mathrm{nl}}$ measured in the standard nonlocal configuration (Fig. 1b) with contact j as injector and i as detector. Hence, in the limit of a low electric field ($E \cong 0$) the 2T SV signal can be approximated by $\Delta V^{E=0} = \Delta V_{\mathrm{S,D}}^{\mathrm{nl}} + \Delta V_{\mathrm{D,S}}^{\mathrm{nl}}$.

In Fig. 3, we compare the measured 2T resistance $\Delta R = \Delta V/I$ (black triangles) with $\Delta R^{E=0} = \Delta V^{E=0}/I$ (blue squares) obtained from the corresponding nonlocal measurements at four injection currents in the linear regime (Supplementary Note 3). The superscript $E = 0$ here means that the detector in the four-terminal nonlocal measurements is unbiased. At a low bias of 0.7 μA, $\Delta R \cong \Delta R^{E=0}$, as expected for negligible electric fields. For higher currents, however, $\Delta R$ substantially exceeds $\Delta R^{E=0}$. The discrepancy increases with bias and at $I = 52\,\mu\mathrm{A}$, the measured $\Delta R$ is 32 times larger than the zero-field value $\Delta R^{E=0}$. This comparison demonstrates that the electric field is indeed responsible for the large MR ratios.

An electric field can affect the MR in two ways. It can (i) alter the transport of $\mu^{\mathrm{s}}$ along the channel or (ii) influence the spin-to-charge conversion. Below, we find that the second effect is dominating here. It has been shown previously that the electric field of a positively (negatively) biased FM/SC tunnel contact can significantly enhance (suppress) the detected spin signal[19, 22–24]. Here, for a positively biased source and a negatively biased drain, this means that $\Delta R_{\mathrm{S,D}}$ is enhanced and $\Delta R_{\mathrm{D,S}}$ is suppressed compared to the low-bias values. The corresponding enhancement and suppression factors can be determined from nonlocal measurements, where an AC spin valve signal is measured in the presence of an additional DC bias across the detector[22, 23] (see sketches in Fig. 3). From such experiments, we find (Supplementary Note 3) that $\Delta R_{\mathrm{S,D}}$ is enhanced by a factor of 4, 30, and 47 for $I = 10$, 30, and 52 μA, respectively, whereas $\Delta R_{\mathrm{D,S}}$ is either unchanged or slightly suppressed.

We also took effect (i) into account by calculating how drift enhances (suppresses) the diffusion of spins from the drain

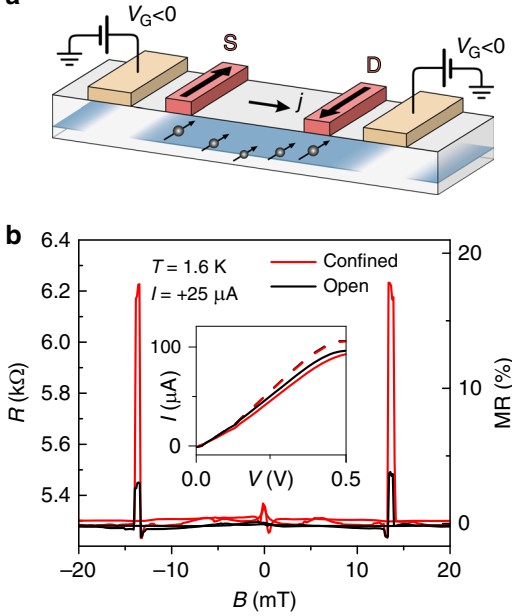

**Fig. 4** Tuning of the MR ratio with gate voltage. **a** Sketch of the spin distribution for AP orientation of magnetization in the confined configuration, i.e., when the 2DES outside the S and D contacts is depleted by applying a negative voltage $V_G$ to the gates. Spin accumulation between the contacts is increased in this case, as spins cannot diffuse outside this region. **b** 2T SV traces for the open channel (black line, $V_G = +1.5$ V) and the confined channel (red line, $V_G = -4$ V). For $I = 25\,\mu$A, the resistance in the antiparallel configuration changes by 14%, corresponding to a voltage change of ~20 mV. Inset: part of the $I–V$ curve in the corresponding bias range. Solid lines correspond to the AP configuration in the open (black) and the confined (red) case. Dashed lines correspond to the P configuration in both cases. Similar measurements were performed on two different devices, giving comparable results (Supplementary Note 5)

(source) to the source (drain). The effect of drift on spin diffusion can be described by modified values of spin diffusion lengths, given by $\lambda_{s,d(u)} = \left[ -(+)\frac{|E|}{2}\frac{\mu}{D_s} + \sqrt{\left(\frac{|E|}{2}\frac{\mu}{D_s}\right)^2 + \frac{1}{\lambda_s^2}} \right]^{-1}$, where $\lambda_s$ is the intrinsic, zero-drift value and $\lambda_{s,d(u)}$ corresponds to the downstream (upstream) spin diffusion length in an electric field $E$[20]. Here, $\mu$ is the electron mobility and $D_s$ is the spin diffusivity. Taking the corresponding experimental values (Supplementary Notes 1 and 3), we find that for $I = +52\,\mu$A, the drift in the channel leads to an enhancement of $\Delta R_{S,D}$ by ~1.4 and a suppression of $\Delta R_{D,S}$ by a factor of two. Therefore, drift along the channel alone has only a minor effect on the two-terminal spin signal. Considering both effects, (i) and (ii), we deduce (Supplementary Note 3) a value for the spin signal at finite electric fields, $\Delta R^{E \neq 0}$ (red dots), which is in good agreement with the measured value $\Delta R$, as shown in Fig. 3.

**Gate-tuning of the MR.** Having demonstrated the drastic enhancement of the 2T MR by increasing the spin-to-charge conversion in our devices, we now address the efficient electrical control of $R$ via control of the spin state, without invoking SOC. We show an alternative scheme of efficiently tuning $R$, based on the control of the diffusion direction of the generated spin accumulation. Blocking the diffusion of spins in the directions away from source and drain by gates (see the sketch in Fig. 4a)

and thus confining the spin accumulation to the region between S and D should result in larger spin signals compared to an open situation, where spins are free to diffuse away from the contacts in both directions[37, 40–42]. Employing gates allows us to switch in a controlled way between open and confined configurations, by depleting parts of the 2DES (Fig. 4a). Indeed, we clearly observe the expected enhancement of the 2T spin valve signal, as displayed in Fig. 4b. It is obtained for a bias current of 25 μA upon switching from the open to the confined configuration. For this current, the MR is increased roughly six times from 3 to 18%. The observed enhancement $\Delta R_{conf}/\Delta R_{open}$ is very close to the value of 5.8 estimated for our device, based on the standard equations, in the limit of low electric fields (Supplementary Note 4). For increased bias values, we find, however, the effect of confinement to be reduced. Similar results were also obtained for the second investigated device (Supplementary Note 5). An understanding of the exact mechanism behind this effect requires further investigations. The results from the gated samples are consistent with control measurements on samples where the confinement was realized by etching parts of the transport channel (Supplementary Note 6).

## Discussion

Combining the electric-field-driven boost of spin-to-charge conversion with the novel concept of gate control of spin accumulation, we have demonstrated a spin transistor-like device with at least one order of magnitude larger MR signals than that reported so far for semiconductor spin valves. The electric field effect which is responsible for the enhanced spin signals has been observed in a number of experiments with different FM/SC contacts, like Fe/GaAs[22], (Ga,Mn)As/GaAs[23], $Co_2MnSi$/GaAs, $Co_2FeSi$/GaAs[24], Fe/MgO/Si[18, 25], and very recently also in two-terminal geometry in graphene[21]. Although specific material properties matter[21, 24], these different systems have in common that the spin signal gets enhanced for positively biased detector contacts. The existing microscopic models explaining such behavior are complex[19] and still not fully understood[24]. Nevertheless, the universal character of the effect suggests that harnessing electric field action could open a pathway toward semiconductor spin transistors with large MR working up to room temperature.

The large signal $\Delta R/R^P$ of up to 20% that we observe in the linear regime is essentially due to the nonrectifying contacts, which allow driving a large current in 2T geometry. The large current generates both, a large spin accumulation and enhances its detection, as discussed above. Further enhancement (up to 80% here) observed in the nonlinear sections of the $I–V$ characteristic originates from the sublinear character of the $I–V$ curve in this regime. When the bias current is applied in the corresponding region (see the blue dashed line in the lower inset of Fig. 2a), the voltage performs a large jump whenever the alignment of the contacts is switched from parallel (black trace) to antiparallel (red trace). Tunnel contacts with $I–V$ characteristics featuring sublinear regions then offer an additional mechanism for the enhancement of MR.

The demonstrated gate-tuning of the spin valve signal, based on the electrical control of spin diffusion in the channel, can be an attractive alternative for SOC-based schemes because of its universality. While other alternatives to SOC-based control[30–33] of spin rely on some specific properties of the given system, electrical lateral confinement should be possible to realize in a wide range of materials, the conductivity of which can be controlled by the field effect. It would be particularly interesting to test this concept in graphene and other novel 2D materials.

## Methods

**Fabrication.** All investigated samples were fabricated from the same wafer, grown epitaxially by molecular beam epitaxy. The wafer consists of the following layers (from top): ferromagnetic $Ga_{0.95}Mn_{0.05}As$ (50 nm), $Al_{0.33}Ga_{0.67}As$ (2 nm), $n^+$-GaAs (8 nm, $n^+ = 5 \times 10^{18}$ $cm^{-3}$), $n^+ \rightarrow n$-GaAs transition layer (15 nm), n-GaAs (100 nm, $n = 6 \times 10^{16}$ $cm^{-3}$), i-GaAs (50 nm), $Al_{0.33}Ga_{0.67}As$ (125 nm), AlGaAs/GaAs superlattice buffer (500 nm), and (001)-oriented GaAs substrate. The top four layers form a spin Esaki diode. The two-dimensional electron system (2DES) is confined at the i-GaAs/$Al_{0.33}Ga_{0.67}As$ interface and charge carriers are provided by a Si δ-doped layer in the $Al_{0.33}Ga_{0.67}As$ region. As the (Ga,Mn)As layer is degenerately p-doped, it is necessary to increase the Si doping in the $n^+$-layer to ensure Esaki diode functionality. This is achieved by the so-called pseudo-δ-doping, meaning that the growth process of the 8-nm-thick $n^+$-GaAs layer using continuous Si flux is stopped every 1.6 nm for 10 s, to accumulate Si dopants. This allows $n^+$-doping higher than $1 \times 10^{19}$ $cm^{-3}$, making the p–n junction more symmetric and thus lowering its resistance. The 10-μm-wide transport channels are defined using photolithography and wet-etching techniques. Narrow, 500-, and 700-nm-wide, ferromagnetic electrodes are defined using electron beam lithography, followed by evaporation of Au/Ti contacts. Different widths of source and drain contacts ensure different switching fields of the contacts during magnetoresistance measurements. In the region between the contacts, the top three layers are partially etched away using wet chemical etching in order to limit the lateral transport to the 2DES. For the samples described in this paper, the etching depth was between 56 and 60 nm. The electric gates were defined using e-beam lithography followed by atomic layer deposition of 40 nm of insulating $Al_2O_3$ and thermal evaporation of 7-nm NiCr. Such gates ensure ~70% transparency to red light ($\lambda = 645$ nm), used to illuminate the samples in order to populate the 2DES.

**Measurements.** Electrical measurements were performed primarily in the local configuration (Fig. 1a), i.e., with the current flowing between the ferromagnetic source and drain contacts, either applying a constant current or a constant voltage. The voltage drop $V_{2T}$ was measured directly between both contacts. SV measurements were performed while sweeping an external magnetic field along the axes of the FM contact stripes between $B = 0.5$ T and $-0.5$ T, values needed to initially saturate the magnetization of the contacts along the external field.

**Data availability.** The data that support the findings of this study are available from the corresponding author upon request.

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

## Acknowledgements

The work has been supported by the German Science Foundation via SFB689.

## Author contributions

All authors contributed to various aspects of the work presented here. M.O. and F.E. fabricated the devices and performed the measurements. T.K. assisted device characterization measurements. M.O. and M.C. analyzed the data. A.B., D.S., and D.B. designed and realized the heterostructure. Project planning was done by M.C. and D.W.; M.C., M.O., D.W., and D.B. discussed the results and wrote the manuscript.

## Additional information

**Competing interests:** The authors declare no competing financial interests.

