## [Peer Review File · Nature Communications]

Reviewers' comments:

Reviewer #1 (Remarks to the Author):

Oltcher et al., report magnetoresistance measurements in GaMnAs/GaAs heterostructures. They observe a large enhancement of magnetoresistance signal due injection bias, which they propose to originate from the electric field effect. Furthermore, they confine the channel by depleting the regions outside the active region and observe an enhancement of the spin-valve signal.

The paper is well written, and the results are reproducible for several devices. The new information provided in the manuscript about this large magnetoresistance signal and its control by bias and confinement will be useful for the research community. However, before recommending publication of manuscript I have following suggestions to improve the manuscript.

- The title with words like Gate-tunable, spin transistor-like, can be misleading, because gate is only used to confine the channel to increase the spin accumulation. Therefore, I suggest to authors to consider a more suitable title according to their results.
- Authors report large spin signals in mV in 2T geometry. Crowell et al., Ref 8, measured mV non-local signal. Dash et al. (Nature 462, 491, 2009) measured mV signal in 3T geometry. These experimental observations of large magnitude of spin signal should be compared with the signals expected with standard spin diffusion model and put into a context.
- Electric bias induced gating effect is also reported by Jansen et al. in Si based 2DEGs (<http://www.nature.com/nmat/journal/v9/n2/abs/nmat2605.html>). It should be discussed in relevant places.
- Recently, Dash group have demonstrated Gate control of spin-valve and Hanle signal in graphene-MoS2 heterostructures at room temperature (<https://arxiv.org/abs/1610.06326>). I suggest authors to cite this paper in introduction and discuss at relevant places.
- Weiss group is well established in spintronics research and I am sure they must have measured Hanle effect in their measured devices. Therefore, I suggest including the non-local Hanle data to the main manuscript. Particularly, interesting things to observe is that, how spin lifetime is affected by electric field from bias current and gate induced confinements.

- If possible, it will be good to discuss about Hanle effect in 2T geometry, as here both the spin accumulation under the contacts and channel contributes to the signal.
- Authors get large MR consistently in the non-linear regime in a narrow bias range. They should write some explanation about this in the manuscript.
- It will be clearer to mention in Fig 2, 3 that measurements are done with 2T geometry.

Minor comments

- Abstract should not have references !
- Citation 20 is missing in the reference list

Reviewer #2 (Remarks to the Author):

Please see the attached review report.

Reviewer #3 (Remarks to the Author):

This manuscript shows the two-terminal magnetoresistance for lateral spin valve geometry in a (Ga, Mn)As/GaAs channel. The authors show the bias dependence of signals and gate control of spins in 2DES with low SOC by switching between uni- and bi-directional spin diffusion. This work is quite interesting, but except gate tuned MR change, other works are not very striking results. Thus, authors should emphasis on the gate tuning part and provide much stronger evidence for gate tuned MR change. To prove the gate induced diffusion blocking and MR enhancement, additional analysis and experiments are strongly required.

The comments on the manuscripts are as follows.

1. I'm not convinced that a large two-terminal MR is indeed spin accumulation data. It is well known that the two-terminal local spin valve measurement is challenging and can be justified with the back-up of the non-local spin generation and detection [Appl. Phys. Lett., 103, 052406 (2013) & Appl. Phys. Lett., 104, 052404 (2014)]. The reasons are the occurrence of strong electrical baseline signals unrelated to the spin accumulation. The authors must improve the discussion about how to separate charge and spin transport when they observe the influence of electric fields on spin-to-charge conversion.
2. As authors mentioned, the large MR in the local spin valve devices was obtained by the

electric field effects at the FM/SC interface (or electric field driven the large spin-to-charge conversion). However, the physical information of the spin-to-charge conversion provided in the manuscript is not enough. In fact, there is no discussion about the charge transport along the channel when they explain the role of electric fields in the large MR in local spin valve devices.

3. In the gate voltage induced confined case, the magnetoresistance noticeably increases. However, the spin signal ΔR also depends on the channel resistance and etc. The author should give clear evidence that the MR enhancement is not from gate induced channel resistance or interfacial resistance but from spin diffusion blocking. Also, gate dependence of channel properties should be addressed.

4. This experiment utilized two gate electrodes in the experiment shown in Fig. 4. Author requires control measurements with an independent control of two gate voltages.

5. The extreme case of diffusion blocking case is that one mill out the outside part of 2DES channel. In this case the channel exists only between the source and drain, so the diffusion blocking may be maximized. This control test should support the author's interpretation.

6. The baseline resistance of all plots should be provided if the plots are shifted.

7. In Figures 2 and 4, there are some peaks at zero magnetic field. Author had better comment on this.

8. In this manuscript, the measurement geometries are placed only in Fig. 1. If the geometry locates near experimental results, readers can understand the data more easily.

9. The cross-sectional view of the channel may be helpful for readers.

In summary, the claims of the large 2T SV MR and the electrical control of the resistance of spin channels with low spin-orbit coupling are interesting. However, there are still many points that authors should address before publishing. So I don't want to recommend it to be published.

Reviewer #2's Comments:

In this manuscript, the authors demonstrate gate tunable large magnetoresistance (MR) in a spin-transistor-like device, which consists of a (Ga,Mn)As/GaAs spin Esaki diode and two-dimensional electron system (2DES). A resistance change up to 80% were observed, and shown to be tuned by gate voltage. More importantly, the gating mechanism is different from the one which utilizes the spin-orbit coupling (SOC), thus *allowing materials with low SOC to be used as the channel of a spi-transistor*. Overall, the work presented in this manuscript is very interesting, and I would like to recommend it to be published in NC after the authors address the following points:

- (1) As can be seen from the I - V curves plotted in Fig. 2(a), a very large MR can only be realized in a small current or voltage range. According to my understanding, this range is restricted to the area covered by the negative differential resistance (NDR), is it possible extend the current or voltage range with high MR?
- (2) Figures 2b and 2c show the typical SV signals measured in the linear and nonlinear regime of the Esaki diode at 1.4 K. In these figures, the coercive field of the two $\text{Ga}_{0.95}\text{Mn}_{0.05}\text{As}$ ferromagnetic contacts with different size is smaller at higher bias current. The authors should comment this phenomenon.
- (3) The high MR seems to require a spin Esaki diode to be contained in the spin transistor device, how to remove this restriction?
- (4) Spin-transistor should finally have the functions of completing logic operations, so it would be great if the authors could demonstrate or conceive several simple logic functions such as “AND” or “XOR” operations based on their devices. Reference “J. Wunderlich *et al.*, Science 330, 1801 (2010)” would be a good example.

Responses to Reviewers:

Referee #1

We would like to thank the Referee for evaluating our paper as “well written” and for finding the results “useful for the research community”. Below we address all her/his comments in detail.

1. The title with words like Gate-tunable, spin transistor-like, can be misleading, because gate is only used to confine the channel to increase the spin accumulation. Therefore, I suggest to authors to consider a more suitable title according to their results.

Answer: As we understand, the Referee is concerned that the title could mislead the readers into thinking that the paper describes the realization of the Datta-Das type of a spin transistor. To avoid this, we replaced the phrase “spin transistor-like device” with “spin valve device”. We prefer, however, to keep “gate-tunable large magnetoresistance” in the title, as we believe it describes properly the content of the paper. The new title is now “Gate-tunable large magnetoresistance in an all-semiconductor spin valve device.”

2. Authors report large spin signals in mV in 2T geometry. Crowell et al., Ref 8, measured mV non-local signal. Dash et al. (Nature 462, 491, 2009) measured mV signal in 3T geometry. These experimental observations of large magnitude of spin signal should be compared with the signals expected with standard spin diffusion model and put into a context.

Answer: We already referred to Peterson et al. (Ref. 8 in the original manuscript, Ref. 24 in the revised one) in our paper, giving that work as an example of the nonlocal signal being enhanced as a result of additional biasing of the detector. The vast topic of spin injection in silicon we addressed mainly through the review paper by Jansen [*Nat. Mater.* **11**, 400–408 (2012).] Following the Referee’s suggestion, in the revised paper we shortly address in the introduction the issue of unexpectedly large spin signals, referring also to the paper by Dash et al. We would like to point out, however, that whereas the spin signals reported in above papers were in the order of single mV, we measured spin signals up to 80 mV in the linear regime and even up to 0.35 V in the non-linear regime. We also did compare those signals with the predictions of the standard spin diffusion model in Fig. 3. There, we “reconstructed” the local signal from nonlocal measurements, following the standard model (see the corresponding discussion in the text and Supplementary Note 3).

3. Electric bias induced gating effect is also reported by Jansen et al. in Si based 2DEGs (<http://www.nature.com/nmat/journal/v9/n2/abs/nmat2605.html>). It should be discussed in relevant places.

Answer: We refer to this work in the Discussion section (Ref. 30), while shortly discussing the alternative to SOC ways of electrical control of the spin signal.

4. Recently, Dash group have demonstrated Gate control of spin-valve and Hanle signal in graphene-MoS2 heterostructures at room temperature (<https://arxiv.org/abs/1610.06326>). I suggest authors to cite this paper in introduction and discuss at relevant places.

Answer: We thank the Referee for bringing this work to our attention. We added the corresponding reference to the introduction and Discussion section (Ref. 33 in the revised manuscript).

5. Weiss group is well established in spintronics research and I am sure they must have measured Hanle effect in their measured devices. Therefore, I suggest including the non-local Hanle data to the main manuscript. Particularly, interesting things to observe is that, how spin lifetime is affected by electric field from bias current and gate induced confinements.

Answer: We did indeed perform nonlocal Hanle measurements on the series of 2D samples, which we discussed in our recent paper [Kuczmik *et al.*, PRB **95**, 195315 (2017)]. We cite this paper in the revised text as Ref. 36. One of the conclusions from those experiments was that low level AC excitation technique is required for reliable measurements of spin relaxation times, in order to limit the influence of dynamic nuclear polarization (DNP) effects on the measured Hanle curves. Large DC currents generate large spin accumulation, which in turn gives rise to a large nuclear field, that strongly narrows the measured Hanle curves near $B=0$. Therefore, quantitative Hanle investigations of electric-field-dependent spin relaxation are quite difficult as the relaxation times come out too long due to DNP. We discuss it shortly in Supplementary Note 7.

6. If possible, it will be good to discuss about Hanle effect in 2T geometry, as here both the spin accumulation under the contacts and channel contributes to the signal.

Answer: We did performed local 2T Hanle measurements both on gated and non-gated samples. We clearly observed suppression of the signal in the out-of-plane field, what confirms that observed large signals are indeed spin-related. We added this information to the main text, while discussing Fig. 2., where we write: *The spin origin of this resistance is confirmed by measurements in an out-of-plane magnetic field, which depolarizes the in-plane spins, thus reducing the measured value of $R^{P(AP)}$ due to suppression of its spin-dependent contribution (Supplementary Note 7).* Unfortunately, for the reason described in the reply to comment 5, i.e., due to strong influence of DNP on the measured curves, it is impossible to extract reliable spin relaxation times from these measurements. We now shortly summarize 2T Hanle measurements in Supplementary Note 7.

7. Authors get large MR consistently in the non-linear regime in a narrow bias range. They should write some explanation about this in the manuscript.

Answer: We would like to stress that MR of $\sim 20\%$ measured in the linear regime (up to $\sim 50 \mu\text{A}$ for the sample in the Fig. 2) can be already considered “large”, as it is at least an order of magnitude larger than reported so far for other semiconductor systems. Non-linearity of the I-V curve is responsible for even further increase of MR, in case of measurements with the constant current bias. We did discuss this issue already in the original manuscript (see discussion of Fig. 2) but in the revised version we even stronger emphasize this point. Particularly in the Discussion section, we devoted one whole paragraph to discuss additional enhancement in the nonlinear region. We write, e.g., : *Further enhancement (up to 80 % here) observed in the non-linear sections of the $I - V$ characteristic originates from the sublinear character of the $I - V$ curve in this regime. When the bias current is applied in the corresponding region (see blue dashed line in the lower inset of Fig. 2a), the voltage performs a large jump whenever the alignment of the contacts is switched from parallel (black trace) to anti-parallel (red trace).*

8. It will be clearer to mention in Fig 2, 3 that measurements are done with 2T geometry.

Answer: We included the corresponding information in the figure captions.

Referee #2

We would like to thank the Referee for finding our paper interesting and for recommending it for publication. Below we address all her/his comments in detail.

1. As can be seen from the I-V curves plotted in Fig. 2(a), a very large MR can only be realized in a small current or voltage range. According to my understanding, this range is restricted to the area covered by the negative differential resistance (NDR), is it possible extend the current or voltage range with high MR?

Answer: We would like to stress that MR of ~20% measured in the linear regime (up to ~ 50 μ A for the sample in the Fig. 2) can be already considered “large”, as it is at least an order of magnitude larger than reported so far for other semiconductor systems. Non-linearity in the I-V curve causes additional enhancement in the case of constant current measurements. For the sample in Fig. 2 the MR reaches 80% in this regime, so this indeed dominates the plot and therefore can lead to the wrong impression that the signal is only enhanced in this narrow bias regime. As already mentioned in the original version of the paper, the difference between the P and the AP branch of the I-V curve leads to the large voltage output (~0.35 V) in the nonlinear regime, for a constant current bias (see the bottom inset in Fig. 2a) . In the revised version we emphasize it even more. We would like to point out here, that it is not the NDR feature that is required, but rather a “flat”, sublinear characteristic corresponding to large differential resistance. We discuss this issue in more details in the revised version of the paper in the Discussion section. We write: *Further enhancement (up to 80 % here) observed in the non-linear sections of the I – V characteristic originates from the sublinear character of the I – V curve in this regime. When the bias current is applied in the corresponding region (see blue dashed line in the lower inset of Fig. 2a), the voltage performs a large jump whenever the alignment of the contacts is switched from parallel (black trace) to anti-parallel (red trace).*

2. Figures 2b and 2c show the typical SV signals measured in the linear and nonlinear regime of the Esaki diode at 1.4 K. In these figures, the coercive field of the two Ga_{0.95}Mn_{0.05}As ferromagnetic contacts with different size is smaller at higher bias current. The authors should comment this phenomenon.

Answer: The coercive field of Ga_{0.95}Mn_{0.05}As decreases typically with increasing temperature, as can be observed both in nonlocal and local measurements. The decrease of the coercive field with current bias, also observed in the injector contact in nonlocal measurements, is a result of the local heating of the contacts. We added a corresponding comment to the caption of Fig. 2 and also show the T-dependence of the spin valve in Supplementary Note 8.

3. The high MR seems to require a spin Esaki diode to be contained in the spin transistor device, how to remove this restriction?

Answer: While the non-linear part of the Esaki diode enhances the spin signal, it is important to note that non-rectifying tunnel contacts are the essential feature to achieve a large MR signal in 2T geometry. Non-rectifying I-V curves enable passing large currents through the pair of opposite-biased contacts, thus enabling to take advantage of electric-field driven enhancement of the spin-to-charge conversion. Non-linearity in the finite bias region leads to an additional increase of the MR in the constant current regime. An Esaki diode constitutes a perfect testbed for studying the above effects as it meets all requirements. One can imagine, however, that other systems can be engineered to produce desired characteristics. We consider this issue in the revised version, where we write in the

Discussion section: *Tunnel contacts with $I - V$ characteristics featuring sublinear regions offer then an additional mechanism for the enhancement of MR.*

4. Spin-transistor should finally have the functions of completing logic operations, so it would be great if the authors could demonstrate or conceive several simple logic functions such as "AND" or "XOR" operations based on their devices. Reference "J. Wunderlich et al., Science 330, 1801 (2010)" would be a good example.

Answer: It would surely be appealing to connect our devices to realize logic operations. Presently, we feel however that it is more necessary to get a better understanding of the bias dependence of the observed effect, as the gating works very well only for low currents flowing between source and drain. Hence, we have so far not made any experimental attempts to realize logic functions as this goes beyond the scope of our present work.

Referee #3

We would like to thank the Referee for finding our work on gate-tuned MR interesting, but we do not agree with her/his assessment, that the large MR signal observed in our devices is not a striking result. Quite opposite, we find it very compelling that we could measure signals orders of magnitude larger than reported so far for semiconducting devices in general and semiconductor based two-dimensional electron systems in particular. We also think some of the issues raised by the Referee were already addressed in the original version of the paper. We discuss it below in details.

1. I'm not convinced that a large two-terminal MR is indeed spin accumulation data. It is well known that the two-terminal local spin valve measurement is challenging and can be justified with the back-up of the non-local spin generation and detection [Appl. Phys. Lett., 103, 052406 (2013) & Appl. Phys. Lett., 104, 052404 (2014)]. The reasons are the occurrence of strong electrical baseline signals unrelated to the spin accumulation. The authors must improve the discussion about how to separate charge and spin transport when they observe the influence of electric fields on spin-to-charge conversion.

Answer: We agree with the Referee that 2T spin valve signals are challenging, therefore we find it quite special that we were able to measure signals larger than in other semiconductor-based systems. In fact, signals reported in our work are two to three orders of magnitude larger than those reported in references given by the Referee. We are aware that the 2T resistance contains spin dependent and spin independent contributions, with the latter being dominated in case of our devices by the resistance of the FM/SC interfaces. We discussed it shortly in the introduction and in the Supplementary Note 1 in the original manuscript and make it more clear in the revised version. We write: *This condition [for large MR] is particularly difficult to satisfy in semiconductors, as a high interface resistance needed for efficient injection increases both the dwell time and spin-independent contribution to $R^{P(AP)}$. This is the reason for small MR < 1% obtained so far for semiconductor channels^{14,16-18} (see Supplementary Note 1).*

It is certainly important to back up 2T results with nonlocal data and this is exactly what we already did in the original version of the manuscript. In the Fig. 3 we showed that the measured large 2T signals can be explained as a result of enhancement of the spin-to-charge conversion at the positively biased contact. We performed nonlocal experiments, where we showed how the AC nonlocal spin signal is enhanced by a positive DC bias applied to the detector. In other words, we “reconstructed” the local signal from nonlocal measurements. All these experiments were described in the Supplementary Note 3 and show that the signal is spin-induced. Thus, we are not sure, what other way of separating spin and charge currents the Referee has in mind?

Nevertheless, in the revised version, we emphasize it even more in the main text. We write in the section *The role of electric fields: Here, for positively biased source and negatively biased drain, this means that $\Delta R_{S,D}$ is enhanced and $\Delta R_{D,S}$ is suppressed compared to the low-bias values. The corresponding enhancement and suppression factors can be determined from nonlocal measurements, where an AC spin valve signal is measured in the presence of an additional DC bias across the detector^{22,23} (see sketches in Fig. 3). From such experiments we find (Supplementary Note 3) that $\Delta R_{S,D}$ is enhanced by a factor of 4, 30 and 47 for $I=10, 30$ and $52 \mu A$, respectively, whereas $\Delta R_{D,S}$ is either unchanged or slightly suppressed.*

Additionally, in the revised version, in the Supplementary Note 7 we show the results of 2T Hanle measurements. We believe that the observed suppression of the signal in the out-of-plane magnetic field will remove any doubts about the spin-origin of the measured large signals. We write now in the

subsection *Magnetoresistance (MR) measurements: The spin origin of this resistance is confirmed by measurements in an out-of-plane magnetic field, which depolarizes the in-plane spins, thus reducing the measured value of $R^{P(AP)}$ due to suppression of its spin-dependent contribution (Supplementary Note 7).*

2. As authors mentioned, the large MR in the local spin valve devices was obtained by the electric field effects at the FM/SC interface (or electric field driven the large spin-to-charge conversion). However, the physical information of the spin-to-charge conversion provided in the manuscript is not enough. In fact, there is no discussion about the charge transport along the channel when they explain the role of electric fields in the large MR in local spin valve devices.

Answer: Although electric field driven large spin-to-charge conversion at FM/SC interfaces has been reported for different systems, the exact mechanism still remains not understood. We discuss it in more details in the Discussion section of the revised paper, where we write: *The electric field effect which is responsible for the enhanced spin signals has been observed in a number of experiments with different FM/SC contacts, like Fe/GaAs²², (Ga,Mn)As/GaAs²³, Co₂MnSi/GaAs, Co₂FeSi/GaAs²⁴, Fe/MgO/Si^{18,25} and very recently also in two-terminal geometry in graphene²¹. Although specific material properties matter^{21,24}, these different systems have in common that the spin signal gets enhanced for positively biased detector contacts. Existing microscopic models explaining such behavior are complex¹⁹ and still not fully understood²⁴.*

If by *the charge transport along the channel*, the Referee means the effect of the drift in the channel on the spin transport along the channel, then we did in fact discuss it in the original version of the paper, where we labeled the corresponding effect with (i). We devoted the whole paragraph to show that it has very limited effect on the measured signals (see discussion of Fig. 3 in the paper and the related Supplementary Note 3) .

3. In the gate voltage induced confined case, the magnetoresistance noticeably increases. However, the spin signal ΔR also depends on the channel resistance and etc. The author should give clear evidence that the MR enhancement is not from gate induced channel resistance or interfacial resistance but from spin diffusion blocking. Also, gate dependence of channel properties should be addressed.

Answer: It is of course true, that ΔR depends on the resistance of the channel between source and drain contacts. Our gates are, however, outside the current path, 2 μm away from the corresponding contacts, and the channel resistance remains unaltered. The exact properties of the 2DES underneath the gates are also rather irrelevant. Important is here only whether the gate is closed or open. This we check by measuring the current flow underneath the gate. If closed by appropriate negative bias, the channel is pinched off and no current can flow. This is actually very similar to the situation proposed by the Referee in point 5.

4. This experiment utilized two gate electrodes in the experiment shown in Fig. 4. Author requires control measurements with an independent control of two gate voltages.

Answer: We performed such experiments for two bias voltages. As expected, closing the channel only on one side enhances the signal less. (see Fig. R1)

Figure R1. Comparison of spin valve signals for the fully open channel (grey line), fully confined channel (red) and partially confined (blue) one in case of **a** Left gate closed, right gate open **b** left gate open, right gate closed. Sketches show the corresponding electric configurations.

5. The extreme case of diffusion blocking case is that one mill out the outside part of 2DES channel. In this case the channel exists only between the source and drain, so the diffusion blocking may be maximized. This control test should support the author's interpretation.

Answer: We performed such experiments on samples with the channel confined by etching out parts of the 2DES by means of reactive ion etching (RIE). Consistently with experiments on gated samples, we observed clear enhancement of the signal in RIE-etched samples, which was smaller for larger bias currents. We did not include those measurements in the original paper. Agreeing with the Referee that readers can find these measurements useful, we summarize them in the revised version in the Supplementary Note 6, referring to them in the main text. We write now: *The results from the gated samples are consistent with control measurements on samples where the confinement was obtained by etching parts of the transport channel (Supplementary Note 6).*

6. The baseline resistance of all plots should be provided if the plots are shifted.

Answer: No baseline resistance was subtracted in either of the plots.

7. In Figures 2 and 4, there are some peaks at zero magnetic field. Author had better comment on this.

Answer: It is commonly known that low-temperature spin injection experiments in III-V materials are strongly affected by dynamic nuclear polarization (DNP) effects [e.g. Salis *et al.* Phys. Rev. B **80**, 115332 (2009)]. They typically appear in spin valve traces around $B=0$. We added one sentence with the corresponding information in the discussion of Fig. 2. (subsection *Magnetoresistance (MR) measurements*) saying that: *At $B=0$ the feature related to dynamic nuclear polarization effects³⁸ is also observed.*

8. In this manuscript, the measurement geometries are placed only in Fig. 1. If the geometry locates near experimental results, readers can understand the data more easily.

Answer: The measurement geometry was placed also in Fig. 4. Following the Referee's suggestion, we added small sketches also to Fig. 3.

9. The cross-sectional view of the channel may be helpful for readers.

Answer: We included the wafer layout in the Supplementary Note 1.

Reviewers' Comments:

Reviewer #1 (Remarks to the Author):

Manuscript

Gate-tunable large magnetoresistance in an all-semiconductor spin valve device

The manuscript shows a very clear methods to achieve large MR signals in semiconductor devices and also it can be tuned.

I am satisfied with the changes made in the revised version of the manuscript and recommend publication in Nature communication.

Reviewer #2 (Remarks to the Author):

I have read the authors' the responses to referees, the revised manuscript and the Supplementary notes carefully. In general, the authors addressed my concerns and other reviewers comments or questions point by point clearly. I would like to recommend it publish in Nature Communications as it is.

Reviewer #3 (Remarks to the Author):

My big concern on the manuscript is that a large 2T MR reported is not resulted from the spin accumulation. I still suspect this issue in spite of the rebuttal given by the authors. Therefore, I personally do not want to recommend it to be published in Nature Communications